



# Differential photosynthetic response of marine planktonic and
benthic diatoms to ultraviolet radiation under various temperature
regimes
Yaping Wu[1], Furong Yue[2], Juntian Xu[2,*], John Beardall[3]
1. College of Oceanography, Hohai University, Nanjing, 210098, China
2. College of Marine Life and Fisheries, Huaihai Institute of Technology, Lianyungang,
222005, China
3. School of Biological Sciences, Monash University, Clayton, Victoria 3800, Australia
* Author correspondence: jtxu@hhit.edu.cn





**Abstract:**

We studied the photophysiological response to ultraviolet radiation (UVR) of two diatoms, isolated from different environmental niches. Both species showed the highest sensitivity to UV radiation under relatively low temperature, while they were less inhibited under moderately increased temperature. Under the highest temperature applied in this study, the benthic diatom *Nitzschia sp.* showed minimal sensitivity to UV radiation, while inhibition of the planktonic species, *Skeletonema sp.*, increased further compared with that at the growth temperature. These photochemical responses were linked to values for the repair and damage processes within the cell; higher damage rates and lower repair rates were observed for *Skeletonema sp.* under suboptimal temperature, while for *Nitzschia sp.*, repair rates increased and damage rates were stable within the applied temperature range. Our results suggested that the response of phytoplankton to UV radiation correlated with their niche environments, the periodic exposure to extreme temperature promote the resistance of benthic species to the combination of high temperature and UV radiation. Furthermore, the temperature-mediated UV sensitivities might also have implications for phytoplankton in the future warming oceans.

Keywords: Diatom, Photosynthetic performance, Temperature, UV radiation




**Introduction**

As the most abundant group of phytoplankton, and one that plays an important role in marine ecosystem function and biogeochemical cycles, diatoms are traditionally divided into centric and pennate species on the basis of their valve symmetry (Round et al., 1990). Centric diatoms are usually, though not invariably, planktonic and pennate species are benthic, and are often found living in different niches (Irwin et al., 2012;Keithan et al., 1988). The distribution of centric diatoms is more widespread, with records for the open ocean as well as coastal water, and they maintain position in the upper mixing layer by maintaining buoyancy with elaborated spines or excretion of heavy ions (Lavoie et al., 2016;Villareal, 1988). In contrast, pennate diatoms are often found in the intertidal zone (Stevenson, 1983). Therefore, the 2 groups of diatom are likely to have evolved different strategies to cope with their niche environments (Barnett et al., 2015;Lavaud et al., 2016;Lavaud et al., 2007).

Temperature affects almost all biochemical reactions in living cells, and is one of the most important factors that determines the biogeography, as well as the temporal variation of phytoplankton (Levasseur et al., 1984). Under global change scenarios, increases in sea surface temperature would re-structure the phytoplankton assemblages in the future ocean (Thomas et al., 2012). At small spatial scales, e.g. the coastal zone, diurnal cycle of tides or meteorological events could expose benthic diatoms to extreme environments, including high PAR and UV exposure as well as larger variations in temperature than found for planktonic species. Hence organisms in such exposed areas should potentially possess highly efficient mechanisms to adapt such environment (Souffreau et al., 2010;Weisse et al., 2016).

In the intertidal zone, UV radiation (UVR) is another driving force. UVR is a component of the solar spectrum, along with photosynthetically active radiation (PAR), and has wide reaching effects on organisms, especially photoautotrophs due to their demands for light energy (Williamson et al., 2014). The penetration of effective UVR in coastal waters is mainly dependent on the properties of the seawater (Tedetti and Sempere, 2006). Previous studies have found that UVR significantly inhibited carbon



fixation by phytoplankton in the surface layer, with less inhibition or even stimulation in deep water due to low UVR and limiting levels of PAR (Gao et al., 2007). Detrimental effects, however, varied seasonally, with less inhibition observed for planktonic assemblages during summer, though UV radiation was the highest. This may be attributable to the higher water temperature which facilitated enzyme-catalyzed repair processes within the cell (Wu et al., 2010). There are few documented studies on benthic species, which actually are potentially more resistant to UVR as they are periodically exposed to high solar radiation during low tide (Barnett et al., 2015).

Photosystem II (PSII) initiates the first step of photosynthesis, converting photons to electrons efficiently, but this complex is very sensitive to light (Campbell and Tyystjarvi, 2012). The subunits of PSII are broken down under UVR or high PAR while repaired by insertion of de-novo synthesized protein (Aro et al., 1993); the repair process eventually reaches a dynamic balance with damage (Heraud and Beardall, 2000). However, these two processes are independent from each other. The photochemical damage is mainly determined by the intensity and spectrum of light (Heraud and Beardall, 2000) and is temperature insensitive, while the repair process is driven by a series of enzyme-catalyzed reactions, and is thus potentially sensitive to temperature changes (Melis, 1999). Previous studies revealed that high temperature alleviated UV inhibition of photosystem II in green algae (Wong et al., 2015), while it interactively decreased photosynthetic activity in microphytobenthos under excessive PAR conditions (Laviale et al., 2015).

Coastal water is a highly productive zone, with most of primary productivity attributed to diatoms (Carstensen et al., 2015). Hence, how diatoms respond to environmental factors, e.g. UV radiation, nutrient pulses or temperature, has been extensively studied (Häder et al., 2011). These responses were often shown to be species-specific, and could correlate with cell size, geometry or distinct mechanisms operated by different species (Halac et al., 2014;Wu et al., 2015). Considering the niches in which diatoms are living, physical and chemical factors are quite different between planktonic and benthic species (Souffreau et al., 2010). In this study, we will



use two isolated species to test the hypothesis that benthic diatoms have a stronger
ability to adapt to potentially stressful solar UV radiation under high temperature
regimes.
**Materials and methods**
1.   Species and culture conditions

98       We collected samples from offshore water and intertidal sediments in the coastal

area of the Yellow Sea. These were re-suspended in seawater, and enriched with Aquil
medium and incubated in a growth chamber for 3 days (Morel et al., 1979). Then a sub-
sample was examined under a microscope, and single cells were picked up with a micro
pipette. *Skeletonema sp.* and *Nitzschia sp.* were chosen for the present study, and were
maintained in Aquil medium in a growth chamber at 15 °C. Prior to the experiment,
both species were inoculated into enriched seawater and cultured semi-continuously in
500 mL polycarbonate bottles, illuminated with cool fluorescent tubes at a photon flux
density of ~200 $\mu mol\ m^{-2}\ s^{-1}$, with a 12:12 light/dark cycle. While temperature was set
at 15, 20 or 25 °C with variation less than 0.5 °C, and culture bottles (triplicates for each
temperature) were manually shaken 2–3 times during light period and randomly
distributed in the growth chamber.
2.   Determination of spectra and growth rate
50 mL of culture was filtered onto a GF/F filter, and extracted in 5 mL absolute
methanol for 2 h at room temperature in a 10 mL centrifuging tube, then centrifuged at
4000 rpm for 15 min (TDZ4-WS, Luxiang Inc.). The supernatant was scanned with a
spectrophotometer (Lambda 35, PerkinElmer) in the range of 280nm-750 nm. The cut-
off filters were scanned in the same wavelength range against air as a blank. Specific
growth rate was estimated from the changes of dark adapted chlorophyll fluorescence,
and calculated as: $\mu = (Ln\ F_2 - Ln\ F_1) / (T_2 - T_1)$, where $F_1$ and $F_2$ represent the steady-
state fluorescence intensity at $T_1$ or $T_2$, respectively.
3.   Experimental set up
The experiments were performed under a customized solar simulator with a 1,000
W xenon arc lamp as the light source. The incident irradiances of UV-B light (280–315





nm), UV-A (315–400 nm), and PAR (400–700 nm) were measured using a broadband radiometer (SOLAR-2UV, TINEL Inc.).

In the middle of the light period, samples of both species in the exponential phase were harvested and directly transferred to quartz tubes (35 mL) at a density of less than 20 µg chl $a$ L$^{-1}$, dark-adapted for 15 min, and milli-Q water (as a control) or lincomycin (final concentration, 0.5 mg mL$^{-1}$, for the determination of damage rate in the absence of repair) were added. The tubes were then placed into a water bath one after another at 1 minute intervals while covered with cut-off filters (ZJB280, ZJB400) that block radiation below 280 or 400 nm, respectively (50% transmission at 280 nm or 400 nm, see Figure A1), to create PAR + UV-A + UV-B (PAB) and PAR (P) treatments respectively. The light levels applied were PAR =440 µmol photons m$^{-2}$ s$^{-1}$ and UVR = 41.6 W m$^{-2}$, while temperature was controlled with a cooling system (CTP3000, Eyela) and was set as the incubation level (termed "acclimated") or incubation temperature +10 °C (termed "short term"), the latter mimicking a moderate increase in temperature in the intertidal zone during a low tide period. After the light exposure, samples were moved into a water bath at the same temperature as light exposure, but under dim light (~30 µmol photons m$^{-2}$ s$^{-1}$), for recovery, effective quantum yields were then measured at 12 min intervals.

4.   Chlorophyll fluorescence measurements

A total of 12 tubes (2 species and 2 radiation treatments) were dark-adapted for 15 min, then each tube was moved into water bath one by one with 1 minute interval for light exposure, and sub-samples were taken to measure the initial chlorophyll fluorescence with an Aquapen fluorometer (AP-C 100, PSI). During the subsequent light exposure, sub-samples were withdrawn every 12 minutes from the quartz tubes for fluorescence measurement, this procedure ensured that every sample was exposed to radiation with exact the same time duration. After five rounds of measurements (60 min), samples that were without lincomycin were transferred into the low light condition under the same temperature for recovery, and chlorophyll fluorescence was measured as above for 60 min.





5. Data analysis

Effective quantum yields were measured with the AquaPen and calculated according to the following equations:

Effective quantum yield = $(F_m' - F_t) / F_m'$

where $F_m'$ is the effective maximal fluorescence, and $F_t$ is the steady-state fluorescence under actinic light.

The relative inhibition of effective quantum yield by UV was estimated according to the following equation:

Relative inhibition (%) = $(P_P - P_{PAB}) / P_P \times 100$,

where $P_P$ and $P_{PAB}$ represent the effective quantum yield under P and PAB treatments, respectively. Relative inhibition was calculated when $P_P$ and $P_{PAB}$ were significantly different.

The rates of UVR-induced damage to photosystem II (PSII) ($k$, min$^{-1}$) were calculated from lincomycin treated samples assuming repair ($r$) under these conditions was zero. Repair rates ($r$, min$^{-1}$) were calculated using non-lincomycin-treated samples with the fixed $k$ values obtained from the parallel experiments with lincomycin. Both calculation were according to the Kok equation (Heraud and Beardall, 2000):

$$\frac{P_t}{P_0} = \frac{r}{k+r} + \frac{k}{k+r} e^{-(k+r)t},$$

where $P_0$ and $P_t$ represent the initial effective quantum yield and yield at time zero and t (minutes), respectively.

The recovery rates under dim light were calculated with a simple exponential rise equation (Heraud and Beardall, 2000) :

$$y = y_0 + c \left(1 - e^{-\alpha t}\right)$$

where y represents the effective quantum yield at time t (minutes) during the dim light incubation, $\alpha$ was the recovery rate, while $y_0$ and c are constants.

Statistical differences among treatments were analyzed with a one-way analysis of variance (ANOVA) and Tukey HSD was conducted for *post hoc* investigation. A confidence interval of 95% was set for all tests.



**Results**



*Skeletonema sp.* had a lower growth rate under 15 and 20 °C (*p*<0.05), while
growth increased significantly and was 23% higher than that of *Nitzschia sp.* under
25 °C (Fig 1) (*p*<0.01). The spectra of methanol extracts of both species had a similar
pattern, *Nitzschia sp.* showed relatively higher absorption in the range of 410-480 nm
under 15 or 20 °C (Fig 2 A, B), and this further increased significantly under 25 °C (Fig
2C). While no obvious peak in the UV range for both species.
The initial photochemical quantum yield of 15 °C grown *Skeletonema sp.* was
around 0.50 during light exposure (incubated under 15 °C), but decreased gradually
toward the end of the radiation treatments, with lower values under PAB than the P
condition (Fig 3A). During the dim light exposure period, the quantum yield recovered
to its initial value within 24 min under P treatment, while PAB treated cells only
recovered partially to ~70% by the end of the dim light incubation (Fig 3A). For 15 °C
grown cells that were incubated under 25 °C, the general patterns were similar as under
15 °C, though with smaller differences between the P and PAB treatments (Fig 3B).
Under dim light, quantum yield of both radiation treatments recovered to near initial
values (Fig 3B). For 15 °C grown *Nitzschia sp.* that was measured at 15 °C, the
decreasing pattern under P or PAB was similar to that of *Skeletonema sp.*, while for
PAB exposed cells, *Nitzschia sp.* could only recover to ~50% of initial value under dim
light (Fig 3C). However, when 15 °C grown *Nitzschia sp.* were incubated at 25 °C for
light exposure, both P and PAB treated cells had higher quantum yields, with less UVR
suppression of photosystem II compared with 15 °C, and PAB exposed cells could
recover to 75% of the initial value when subsequently incubated under dim light (Fig
3D).
The 20 °C grown *Skeletonema sp.*, independent of incubation temperatures (20 or
30 °C), showed insignificant UV inhibition for most of time points during radiation
exposure, and recovered more quickly under dim light, especially for PAB treated cells
compared with samples under 15 °C (Fig 4 A, B). For *Nitzschia sp.* that were grown at



20 °C, cells showed moderate UV inhibition during radiation exposure, and the
quantum yield under PAB treatment only recovered to ~80% at the end of the dim light
incubation at 20 °C, while quantum yield recovered to the initial value in cells measured
under 30 °C (Fig 4 C, D).
*Skeletonema sp.* that was grown and measured at 25 °C showed a similar pattern
to that grown under 20 °C during both radiation exposure and subsequent dim light (Fig
5A). However, quantum yields decreased significantly once cells were moved into
35 °C, with much lower values observed under PAB and P treatment ($p<0.001$) than
under 25 °C. During the dim light period, *Skeletonema sp.* only recovered to ~30% for
P treatment, while there was no recovery after the PAB treatment (Fig 5B). For
*Nitzschia sp.* measured under 25 or 35 °C, both treatments showed a similar response,
with lower values under PAB than P during the radiation exposure ($p<0.001$ at 25 °C,
$p<0.01$ at 35 °C), while cells could recover to near initial values at the end of the dim
light incubation (Fig 5 C, D).
In the presence of lincomycin, changes in effective quantum yield showed a
similar pattern for most of treatments (Figure A2-4), except for *Skeletonema sp.*
incubated under 35 °C, which had relatively lower values compared with samples under
25 °C (Figure A4).
The relative inhibition induced by UV radiation at the end of radiation exposure is
shown in Fig 6. Both species had the greatest sensitivities under 15 °C, with 80% and
70% relative inhibition of photochemical quantum yield for *Skeletonema sp.* and
*Nitzschia sp.*, respectively. In the range of acclimated temperature, relative UV
inhibition decreasing with increase of temperature for both species. While in the range
of short term incubation with a 10 °C increase, UV inhibition of *Skeletonema sp.* was
comparable at 25 °C and 30 °C, but increased significantly to ~50% at 35 °C ($p<0.01$).
For *Nitzschia sp.*, relative UV inhibition during short term incubation reached a plateau,
in the range of 25 – 35 °C, of around 25%.
During radiation exposure, the repair rates for photosystem II in *Skeletonema sp.*
varied among different temperatures, with highest values observed at 25 °C, and lowest



values at 35 °C for both radiation treatments (Fig 7A). The damage rates gradually decreased from 15 to 25 °C, then increased significantly toward 35 °C (Fig 7B) ($p<0.001$). The ratio of repair rate to damage rate ($r : k$) showed a unimodal pattern with peak values at 25 °C, and with lowest values under 15 or 35 °C, especially for the PAB treatment (Fig 7C).

The repair rate during light exposure for *Nitzschia sp.*, increased significantly in the temperature range of 15 to 25 °C ($p<0.001$), while kept relatively stable from 25 to 35 °C (Fig 8A). The damage rates were quite stable for all temperatures tested, whether cells were acclimated or exposed to short term elevation of temperature, with mean values around 0.075 for PAB and 0.032 for P treatment (Fig 8B). The $r : k$ ratio increased with temperature in the range of 15-25 °C, reaching relatively stable values of around 1.50 for PAR, and around 1.0 for the PAB treatment (Fig 8C).

Under dim light, the rate constant for recovery of PAR-exposed *Skeletonema sp.* were around 0.10-0.15 min$^{-1}$ in the range of 15-30 °C, while increased significantly to around 0.30 at 35 °C ($p<0.01$) (Fig 9A). The rate constant for recovery of P exposed *Nitzschia sp.* was relatively stable, around 0.25 min$^{-1}$, in the range of applied temperature (Fig 9B). The rate constant for recovery of PAB exposed *Skeletonema sp.* showed an increasing pattern from 0.05 to 0.17 min$^{-1}$ in the range of 15-25 °C, but decreased significantly at 30 °C ($p<0.05$); at 35° values were unable to be estimated due to poor fitting of data points (Fig 9C). No consistent trend was found for the rate constant for recovery of PAB exposed *Nitzschia sp.,* around 0.10-0.15 min$^{-1}$, in the range of applied temperature (Fig 9D).

**Discussion**

The natural variation of physical and chemical factors, including nutrients, salinity, temperature, light etc., provide major controls that determine the distribution, succession and composition of phytoplankton (Levasseur et al., 1984). In response to these variables, phytoplankton have evolved different strategies of acclimation or adaptation (Irwin et al., 2015;Padfield et al., 2016). In this study, we found that both benthic and planktonic diatoms were less inhibited by UVR under moderately increased





temperature, while the benthic species was more resistant to UVR under the extreme
temperature. These findings imply that temperature is a key factor that mediates the
response of diatoms to UVR, while different species have developed distinct
mechanisms in response to their particular niche environments (Laviale et al., 2015).
As a basic environmental factor, temperature affects all metabolic pathways, and
extreme or sub-optimal conditions are often encountered by various organisms in nature
(Mosby and Smith, 2015). The growth response of phytoplankton to temperature varies
from species to species, but often shows a unimodal pattern (Brown et al., 2004;Chen,
2015). For the applied temperature range in the present study, the growth rate of benthic
species showed a slight response, while growth increased with temperature to a greater
extent in the planktonic species, particularly above 25 °C. However, life forms in the
natural environment are affected by multiple stressors concomitantly (Boyd et al., 2015).
For instance, a recent studies have demonstrated that increased temperature would
interactively affect phytoplankton with light intensity (Edwards et al., 2016), and could
alleviate UV direct inhibition on some sensitive species (Halac et al., 2014). When
species were acclimated under sub-optimal temperature (15 °C), both showed obvious
sensitivity to UVR (Fig 3). During the recovery period, the effective quantum yield of
the benthic diatom could rapidly reach the highest values within 12 min irrespective of
the incubation temperature. The planktonic diatom, however, only performed better
under short term elevated temperature. This suggests that the benthic species could have
broader adaptability in cope with the highly varied temperature environment they
frequently experience (Laviale et al., 2015).
The operation of Photosystem II is sensitive to light intensity as well as quality.
High P and UVR can usually induce significant damage to this complex, while the de
novo synthesis of protein can replace the damaged subunit (Aro et al., 1993;Lavaud et
al., 2016). The damage rate ($k$), which represents the efficiency of detrimental effects,
showed a different response for the 2 species in this study; in the planktonic species, $k$
was sensitive to temperature change with the lowest value at the medium temperature,
but was quite stable in the   benthic species at all temperatures tested. This could be



attributed to a decrease in electron transport, or changes in ultra-structure which
resulted in higher intracellular light exposure for planktonic species (Melis, 1999;Nitta
et al., 2005). The repair rates ($r$) and the ratio of $r$ to $k$ further demonstrated that the
planktonic species had a relatively lower optimal temperature in response to UVR, with
the highest $r{:}k$ and lowest UV inhibition at 25 °C. In contrast, in the benthic species $r$
and $r{:}k$ increased steadily and reached relatively stable values at the highest temperature,
and this coincided with lower UV inhibition, implying that although acclimated in lab
condition for weeks, this species still had an active mechanism to respond to high
temperature and UVR, as might occur in its natural niche environment (Laviale et al.,

304 2015).

In addition to repair process that are initiated after damage, UV absorbing
compounds could directly screen out part of the detrimental radiation, protecting
cellular organelles from UV damage (Garcia-Pichel and Castenholz, 1993). In diatoms,
however, the spectra of methanol extracts showed only a small absorbance peak in the
UVR. Unlike xanthophyll cycle related pigments, UV-absorbing compounds (UVAC)
are inducible and only synthesized under long-term UV exposure, indicating that UVAC
are not a major protecting mechanism for lab cultured diatoms (Helbling et al., 1996).
However, the xanthophyll cycle could respond quickly under photo-inhibition, and has
been shown to be a major mechanism in diatoms in response to high light or UV
(Cartaxana et al., 2013;Zudaire and Roy, 2001). Therefore, the relatively higher
absorption in the blue range for benthic species, might indicate that temperature
enhances the synthesis of xanthophyll related pigments (Havaux and Tardy, 1996).
The temperature dependent response to UVR has major implications for
phytoplankton. With the continuing emission of greenhouse gases, the surface seawater
temperature is predicted to increase by up to 4 °C by the end of this century (New et al.,
2011), and this could potentially re-shape the phytoplankton assemblages (Thomas et
al., 2012). While the situation might be more complex in the natural environment with
the consideration of interaction of UVR with other factors (Beardall et al., 2009), for
unicellular green algae, an increase of temperature could mitigate UVR harm for





temperate species, while exacerbating UV inhibition for polar species (Wong et al.,
2015). Moreover, the tolerance of phytoplankton to extreme temperature would be
latitude dependent; for tropical areas where the temperature is already high, an increase
of temperature reduced the richness of phytoplankton (Thomas et al., 2012).

328       The present study showed a differential response to UV radiation for two diatoms

from contrasting niches. As predicted, the benthic species had a higher tolerance to the
combination of extreme temperature and UV radiation, which can be attributed to the
environment in which were living. Below the optimal temperature, both species
performed better in response to UV radiation under elevated temperature, suggesting
that the natural variation of temperature due to changes in the heat flux from the sun or
meteorological events would alter the extent of UV effects on primary producers, and
therefore the aquatic ecosystem (Häder et al., 2011). Furthermore, considering the
projected global warming scenarios, UV radiation could impose different impacts on
phytoplankton with respect to the regional differences (Beardall et al., 2009;Xie et al.,

338    2010).

*Acknowledgement:* This study was supported by National Natural Science
Foundation of China (41476097) and the Fundamental Research Funds for the Central
Universities (2016B12814).




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





**Fig legends:**

Fig 1 The specific growth rates of both species under different temperature levels, vertical lines represent SD, n=3.

Fig 2 The absorption spectra of methanol extracts of *Skeletonema sp.* and *Nitzschia sp.* cultured under different temperature, spectra were normalized with value set as 1.0 at wavelength of 665nm, vertical lines represent SD, n=3.

Fig 3 The quantum yields of 15 °C grown *Skeletonema sp.* and *Nitzschia sp.* under P or P+UVR for 1 hour exposure and subsequent recovery under dim light (gray area) for 1 hour, that were incubated and measured at 15 °C (A: *Skeletonema sp.*, C: *Nitzschia sp.*) or 25 °C (B: *Skeletonema sp.*, D: *Nitzschia sp.*) , vertical lines represent SD, n=3.

Fig 4 The quantum yields of 20 °C grown *Skeletonema sp.* and *Nitzschia sp.* under P or P+UVR for 1 hour exposure and subsequent recovery under dim light (gray area) for 1 hour, that were incubated and measured at 20 °C (A: *Skeletonema sp.*, C: *Nitzschia sp.*) or 30 °C (B: *Skeletonema sp.*, D: *Nitzschia sp.*) , vertical lines represent SD, n=3.

Fig 5 The quantum yields of 25 °C grown *Skeletonema sp.* and *Nitzschia sp.* under P or P+UVR for 1 hour exposure and subsequent recovery under dim light (gray area) for 1 hour, that were incubated and measured at 25 °C (A: *Skeletonema sp.*, C: *Nitzschia sp.*) or 35 °C (B: *Skeletonema sp.*, D: *Nitzschia sp.*) , vertical lines represent SD, n=3.

Fig 6 The relative inhibition induced by UVR on the photosystem II of *Skeletonema sp.* (A) and *Nitzschia sp.* (B) under grown or short term elevated temperature, vertical lines represent SD, n=3.

Fig 7 The repair rate (A) and damage rate (B) of photosystem II in *Skeletonema sp.* during P or P+UVR exposure under grown temperature (acclimated) or short term elevated temperature (short_term), and the ratio of repair to damage rate (C), vertical lines represent SD, n=3.

Fig 8 The repair rate (A) and damage rate (B) of photosystem II in *Nitzschia sp.* during P or P+UVR exposure under grown temperature (acclimated) or short term elevated temperature(short_term), and the ratio of repair to damage rate (C), vertical lines represent SD, n=3.

Fig 9 The rate constants for recovery of P exposed *Skeletonema sp.* (A) and *Nitzschia sp.* (B), and rate constants for recovery of PAB exposed *Skeletonema sp.* (C) and *Nitzschia sp.* (D) under dim





497    light, samples were incubated under grown temperature (acclimated) or short term elevated

498    temperature (short_term), vertical lines represent SD, n=3.



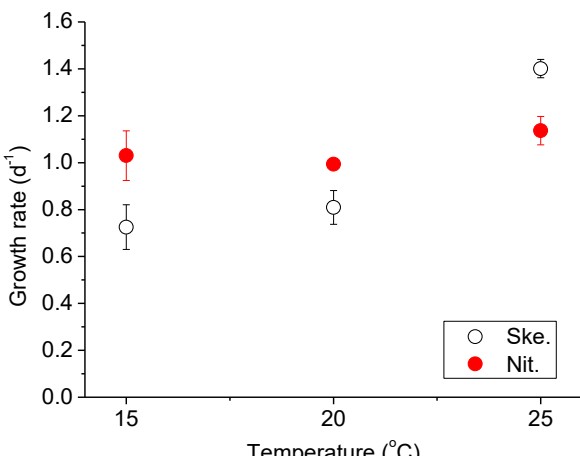

Fig 1

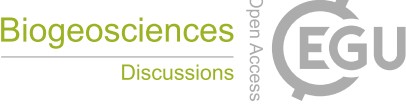



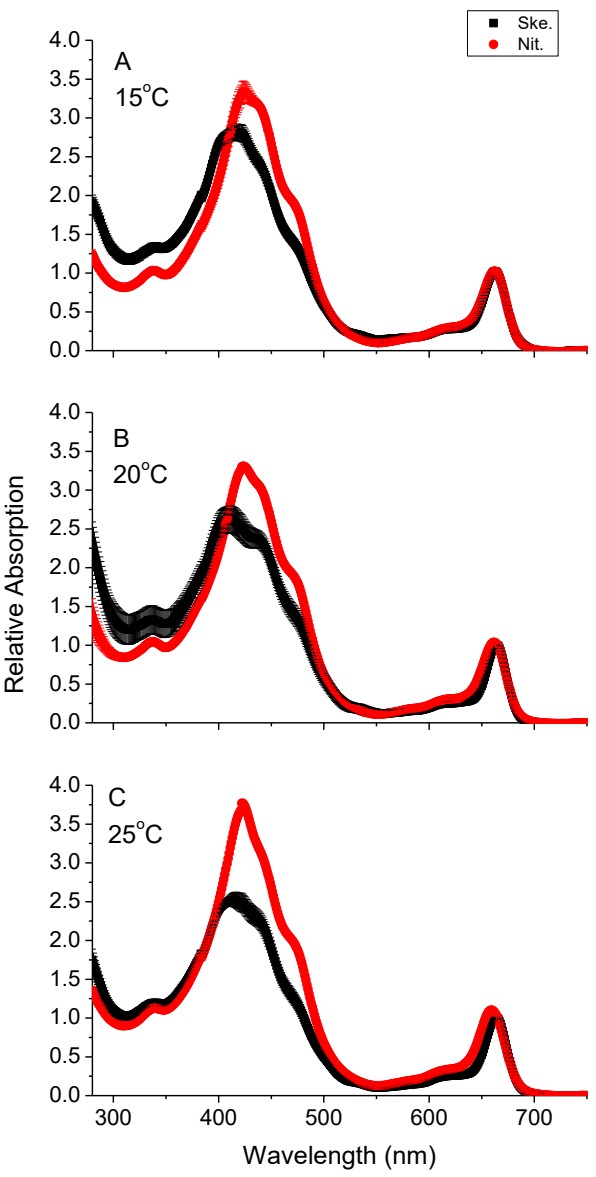

Fig 2





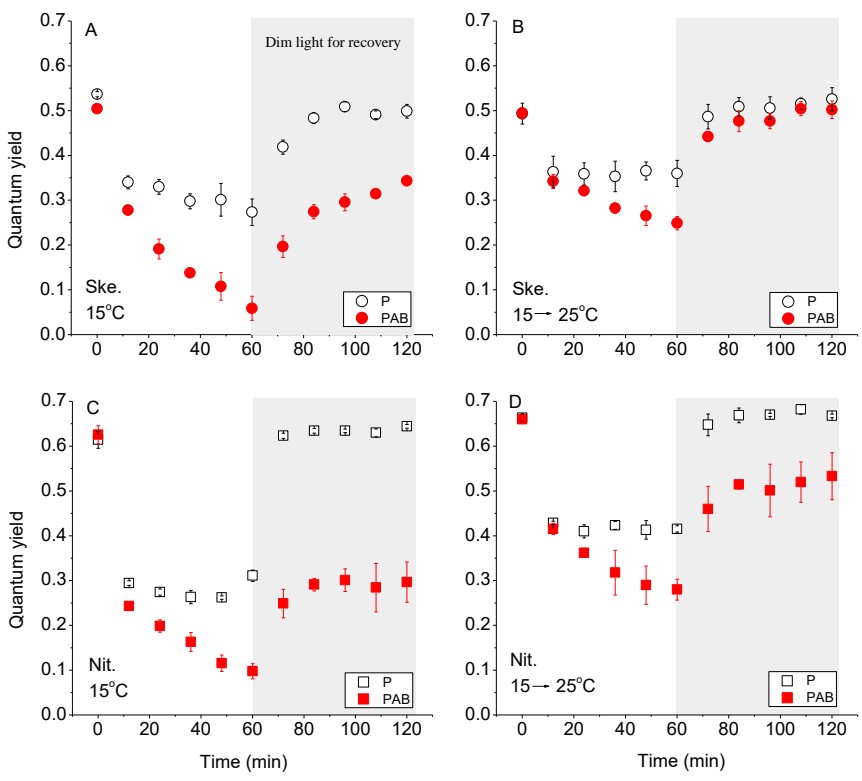

Fig 3





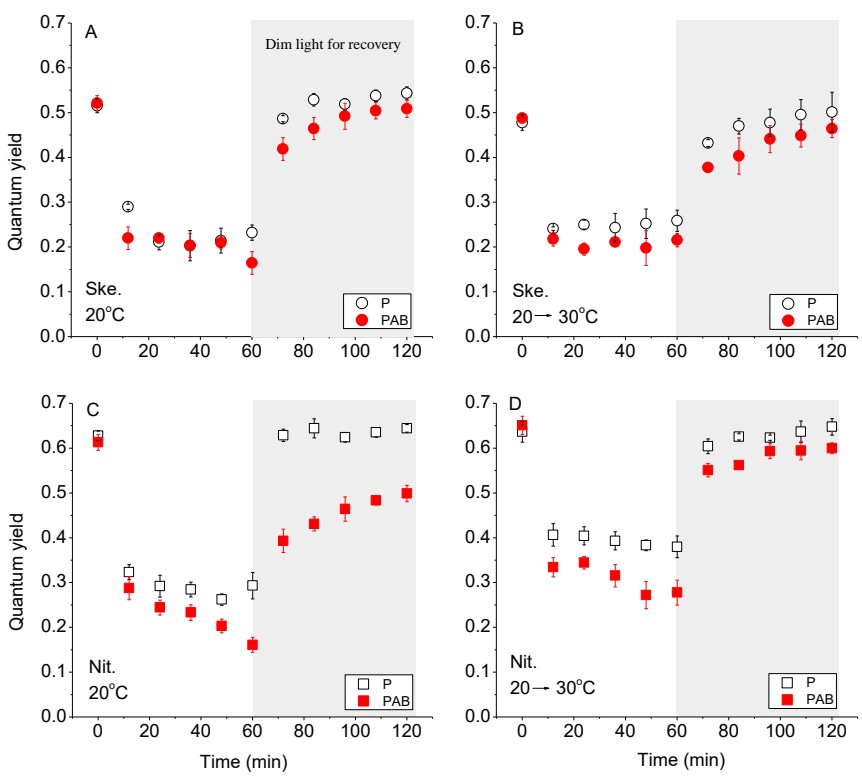

Fig 4





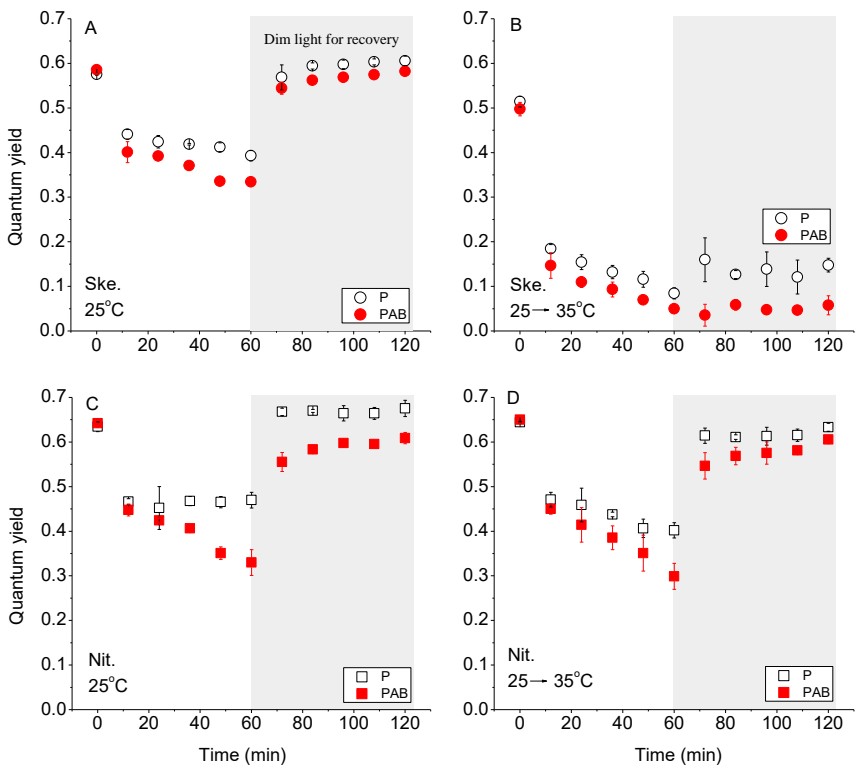

Fig 5




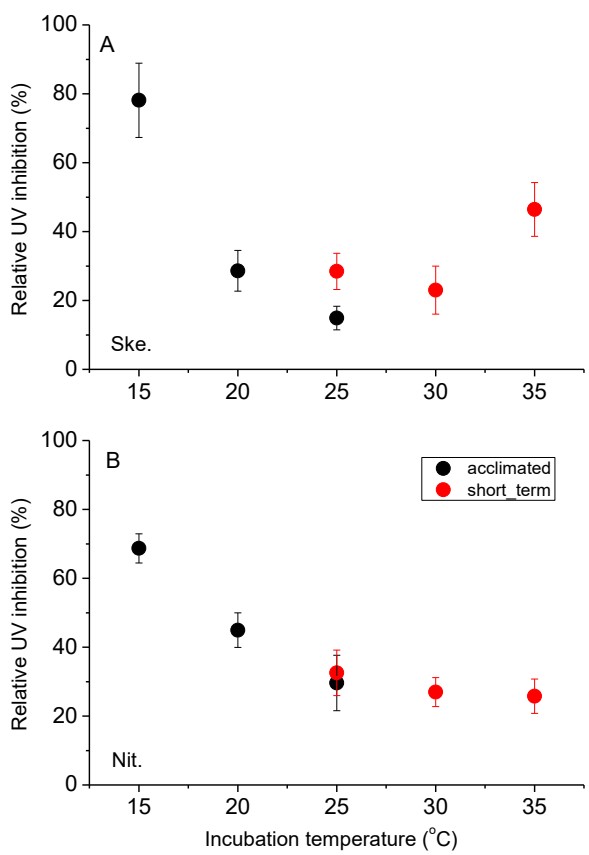

Fig 6




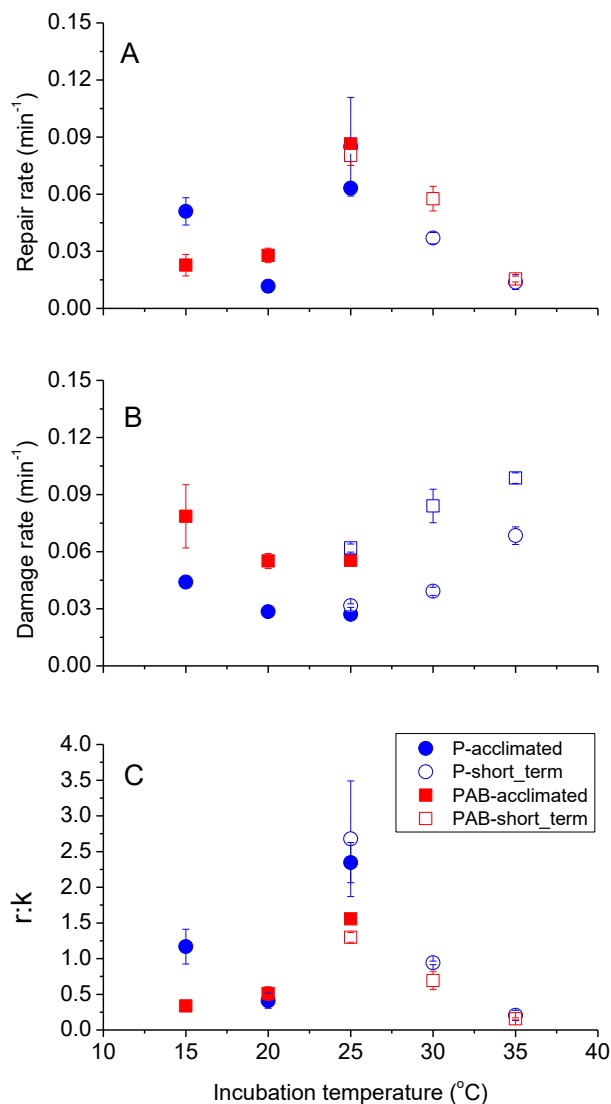

Fig 7





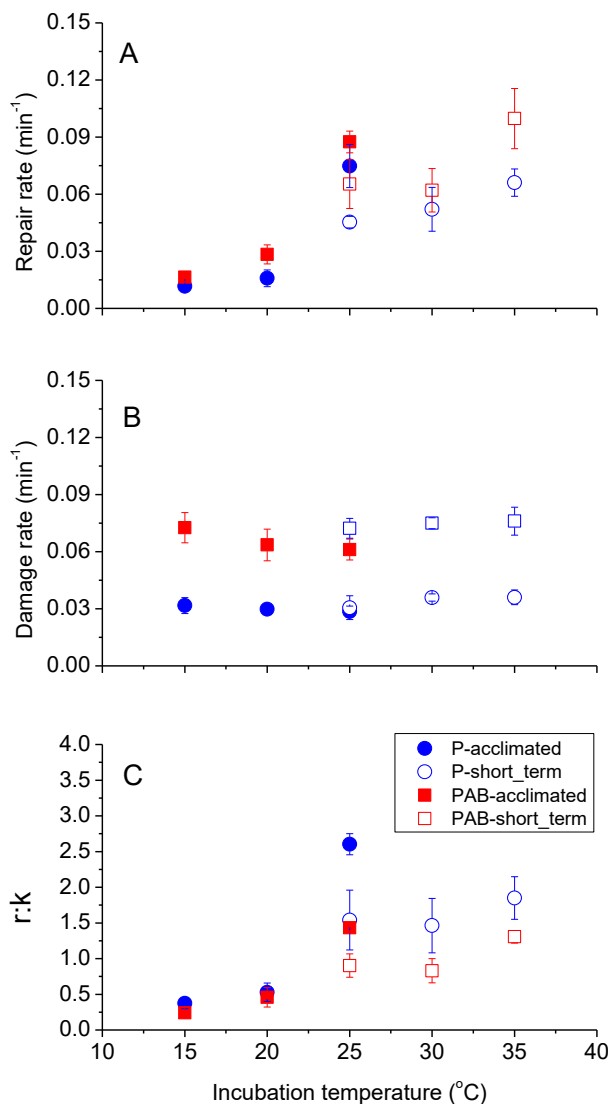

Fig 8



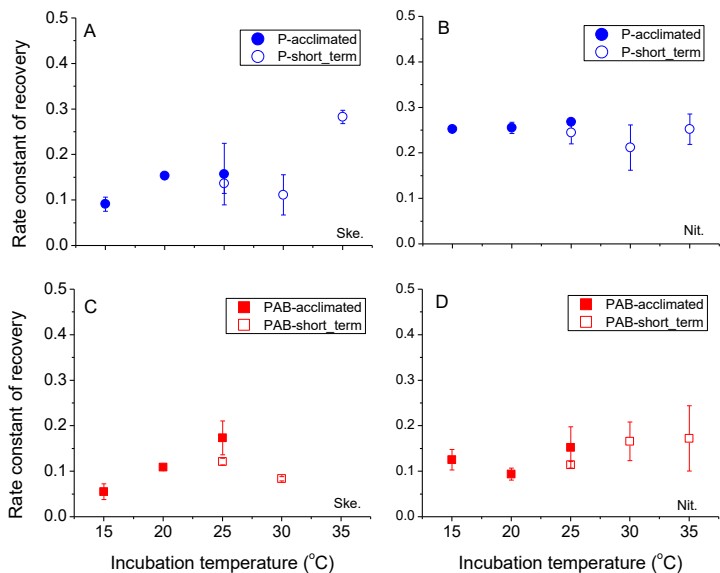

Fig 9



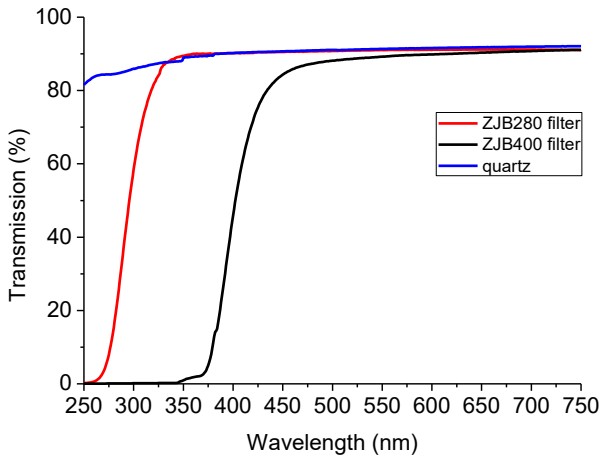

Figure A1The transmission spectra (in percentage) of different cut-off filters (ZJB280, ZJB400) and the quartz tube between 280 and 750 nm.





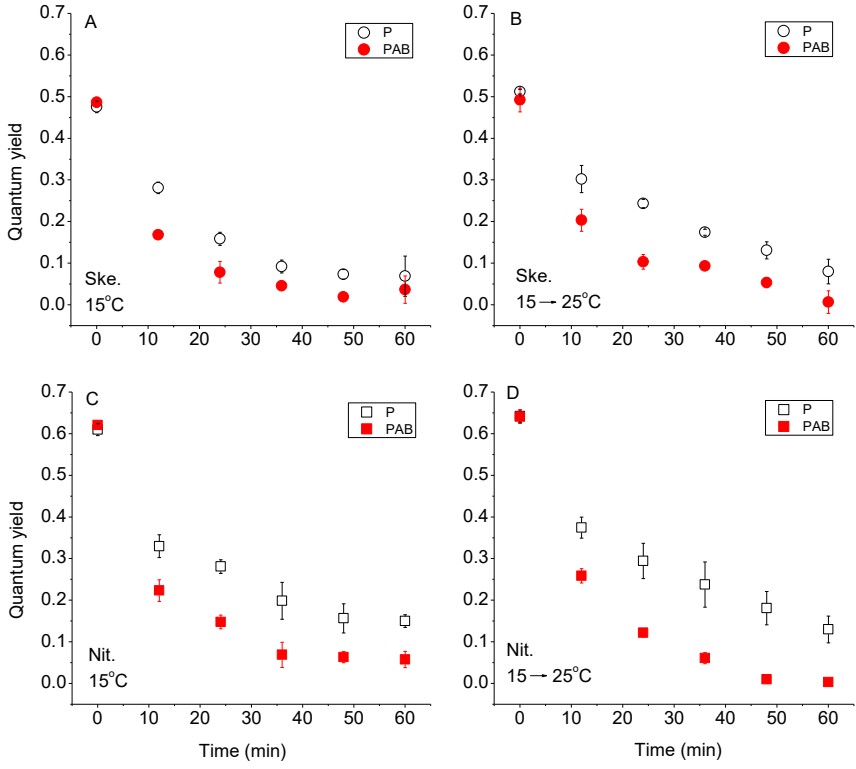

Figure A2 The quantum yields of 15 °C grown *Skeletonema sp.* and *Nitzschia sp.* under
P or P+UVR for 1 hour exposure in the presence of lincomycin, that were incubated and
measured at 15 °C (A, C) or 25 °C (B, D) , vertical lines represent SD, n=3.





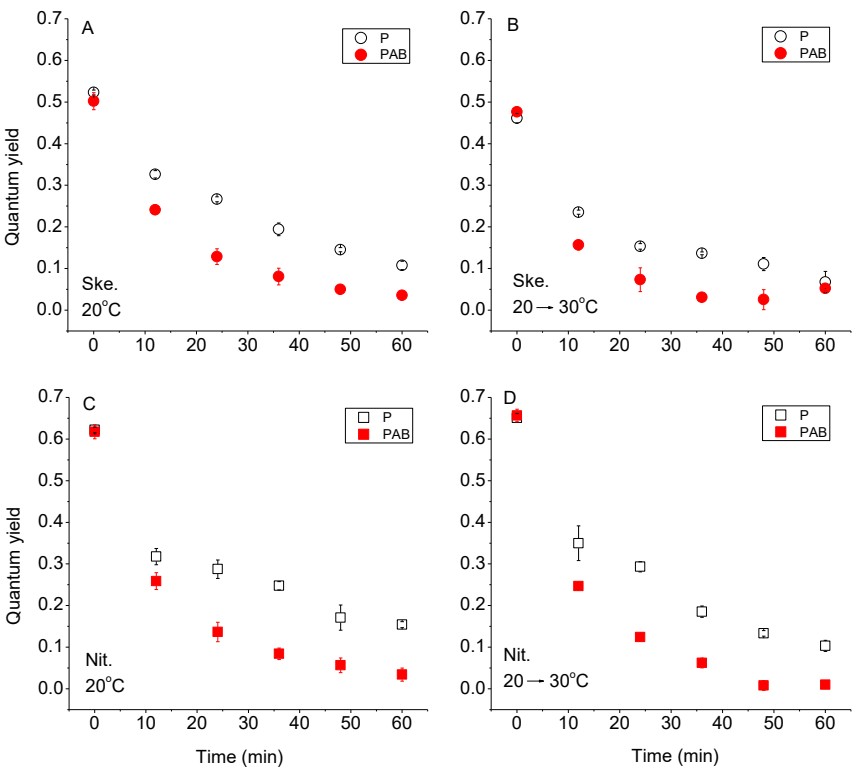

Figure A3 The quantum yields of 20 °C grown *Skeletonema sp.* and *Nitzschia sp.* under
P or P+UVR for 1 hour exposure in the presence of lincomycin, that were incubated and
measured at 20 °C (A, C) or 30 °C (B, D) , vertical lines represent SD, n=3.





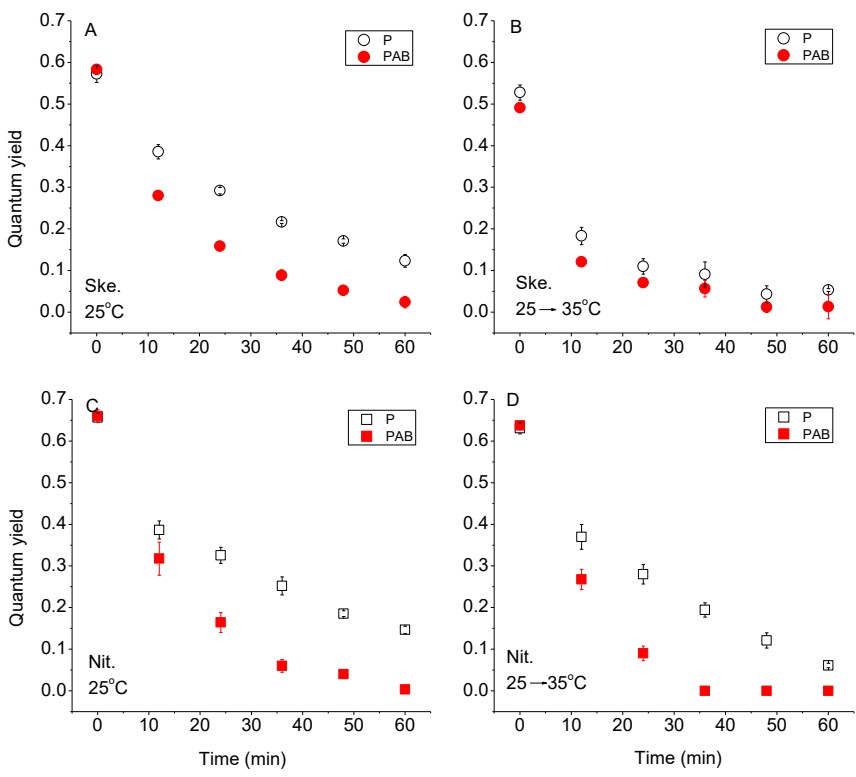

Figure A4 The quantum yields of 25 °C grown *Skeletonema sp.* and *Nitzschia sp.* under
P or P+UVR for 1 hour exposure in the presence of lincomycin, that were incubated
and measured at 25 °C (A, C) or 35 °C (B, D) , vertical lines represent SD, n=3.