# Peer review of "Differential photosynthetic response of marine planktonic and benthic diatoms to ultraviolet radiation under various temperature"

_Biogeosciences, 2017_

## Referee Comment (RC1) · Anonymous Referee #1 · 16 Apr 2017

**Differential photosynthetic response of marine planktonic and benthic diatoms to ultraviolet radiation under various temperature regimes**

Yaping Wu[1], Furong Yue[2], Juntian Xu[2,*], John Beardall[3]

bg-2017-76-manuscript-

**General comments**

This paper presents results from laboratory experiments manipulating the UVR (two levels) and temperature to assess the sensitivity of two diatom species to both factors. The experiment was performed during 120 minutes a single time. The study deals with an interesting topic to phytoplankton ecologists, and tries to clarify a relevant question on the differential photosynthetic responses of the benthic and planktonic species in coastal areas against a scenario of global warming. However I find several problems in the manuscript:

The first impression after reading this manuscript is that it is rather long for the type of study done. The topic is interesting, but this is really a snapshot experiment on two hours on two diatoms species. The most suitable presentation of these results would be/could be as a Note and not as a full length paper. On the other hand, this very short-term experiment, with increments of 10 C in temperature, is very unrealistic. Furthermore, a conclusion like this; "the temperature-mediated UV sensitivities might also have implications for phytoplankton in the future warming oceans" seems to me too much speculative.

My main concern is related to the statistical analysis performed in this study which is not suitable to the experimental design performed and to test the working hypothesis. The authors manipulated two independent factors, so they should do a two-way ANOVA.
Also, when authors analysed the effect on variation in the time of the photosynthetic response to light and dim, they should use a RM-ANOVA. Only when they evaluated the temperature effect on the relative UVR inhibition (%), one-way ANOVA is the correct statistical procedure. Moreover, to test their hypothesis, the authors should evaluate the interactive effect UVR and temperature on the two species as well as to quantify the magnitude of these interactive effects. To my impression a wrong test was used. This fatal error determines that the results and discussion must be re-written.

The estimation of the growth rates is confusing. From the description done, it is not easy to understand how was calculated. If I have understood, it was calculated on fluorescence variation in a 1-hour interval of time, so unit cannot be day; Moreover, I think that the fluorescence is not a good proxy of biomass or abundance, therefore these values did not represent an accurate measurement of growth rates; caution should be taken to discuss this result with those from literature generally obtained from changes of biomass or abundance.

In the results section, there is a lack of precision in the description of the results, making them difficult to understand. The authors should consider remove some of the figures (e.g. Fig. 1 and Fig 2). I think that the figures should be regrouped in two panels, one per each specie, it could benefit the understanding of the Ms. You should present the results in a more synthetic way.

I would like to see the results of the statistical analysis in tables, with the df, F and p values. Likewise, the post hoc results should be presented as part of the figures (lowercase letters).

The authors should pay attention to repetition through the text of terms which was defined in M&M (for instance, photosystem II (PSII), damage rate (k) repair rate (r), Effective quantum yield (y) etc… Likewise, the authors should be consistent with the name of treatments (P-exposed not PAR-exposed; UVR vs PAB ) through the text; and in figure legends the radiation treatments are written as P or P+UVR whereas in graphs are shown as P and PAB. Finally, the variables should be clearly defined, ( e.g. Relative UV inhibition (%) in figures but in line 159 Relative inhibition (%) etc…).

**Specific comments**

**Abstract**

It is Ok

**Introduction**

Line 85-90. This paragraph might seems repetitive.

**Method:**

Using the Aquapen fluorometer the authors had to remove 4 ml for each measurement ( I'm assume that the cuvette is 1 cm ), there are 5 measurements in light, 5 in dim plus an initial sample, so in sum about 45ml are needed. How this work if the sample volume had only 35ml?. This needs to be clarified.

Line 104. *both species were inoculated into enriched seawater*… It would be necessary to give more details about the culture medium, please.

Line 110. *Determination of spectra*, What do you mean?

Line 114 . This sentence *The cut- off filters were scanned in the same wavelength range against air as a blank.* I think it is not the suitable place, because it makes the text confusing.

Line 141. A total of 12 tubes (2 species and 2 radiation treatments)…..? The temperature treatments were not made simultaneously? Moreover, how were done the measured of acclimated vs. short-term samples? I can´t understand how the experiment was performed. I hope to be wrong, but seems that the experiment was not a full factorial. In my opinion, the paper would benefit if an illustration of the experimental design would be included.

Line 169. This sentence "*where $P_0$ and $P_t$ represent the initial effective quantum yield and yield at time zero and t (minutes), respectively*" is confusing, perhaps is better …… where $P_0$ and $P_t$ represent the effective quantum yield at time zero and t (minutes), respectively.

The propagation errors should be applied to calculate the variance of the relative inhibition UVR (as percentage) as well as the variance in the quotient r:k

**Results**

Lines 181-186. This paragraph should be removed because the data are not very informative

.Line 222-225. I'm sorry, but I don't reach to see what brings to this study the treatments with antibiotic.

This section presents comparisons among different temperatures and radiation treatments which could not be evaluated by one-way ANOVA, and post hoc analysis, except to the relative inhibition UVR variable. See above

**Discussion**

Line 260-264. This paragraph is very general; I would like to read something about what is the main contribution of this study.

The discussion, probably will be modified after addressing the points and questions related with experimental set-up and statistical analysis.

---

## Short Comment (SC1) · 7 May 2017

First of all, thanks for the comment. I would like to respond to the major criticisms shortly, before the final response. 1. The reviewer pointed out that 10 degree C increase is very unrealistic. However, the present manu is focusing on the temperature increase on the intertidal flat during low tide period, rather than a mimic of future scenario of global warming. As measured by Laviale et al., (2015, Environmental Microbiology), the in situ temperature change on the intertidal flat was greater than 10 degree C. So I felt that the simulation of temperature is close to what happened in the nature environment. I realized that the last sentence in the abstract might confuse the reviewer that we are dealing with global warming issue, I will delete it in the revision. 2. I agree

with the reviewer that some of data should be analyzed by RM- and two-way ANOVA. I have done this statistical work here, and will incoporated these results to the revision. The major conclusions would not change, while some of the intepretation need to be revised. 3. For the results, Fig 1 and Fig 2 will be removed to supplimentary, as suggested by the reviewer. 4. For the Aquapen fluorometer, the full volume of cuvette is around 4ml, while during the experiment, we withdrawed 2ml for measurement (which is adequate by pre-test). So 35ml sample is enough for the whole experiment. 5. For the propagation errors, we calculated all data with the original Fv/Fm of triplicate, e.g. UVR inhibition, repair, damage rate, and r:k ratio. So the errors were not propagated during the calculation. 6. For the edtorial comments, we will revise the text accordingly.

Dr. Yaping Wu Associate Professor, College of Oceanography, Hohai University.

―――――――――――――――――

---

## Referee Comment (RC2) · P. Neale (Referee) · 28 Jun 2017

Review of Differential photosynthetic response of marine planktonic and benthic diatoms to ultraviolet radiation under various temperature regimes
Author(s): Yaping Wu et al.
MS No.: bg-2017-76
MS Type: Research article

Wu et al. present a study of the photophysiological responses of two diatoms as affected by the temperature during exposure.  The responses are observed during short-term exposures to high light (with and without UV) and subsequent recovery periods in low light.  By tracking the kinetics of PSII quantum yield during the treatment, inferences can be made about the relative contribution of damage and repair processes to the variations in response between temperature.  Additional information can be obtained by exposing the diatoms in the presence of the repair inhibitor lincomycin.  This type of approach has been in previous studies of how variation in environmental factors influence inhibition and recovery kinetics, however most studies have focused on a single time scale of treatment, usually on the order of hours to a few days.  This study is distinctive in comparing the response to a short-term increase in temperature to responses for cultures acclimated over some growth period to the same temperature.  One detail that should be added, however, is how long the acclimated cultures were maintained at their growth temperature before the experiment.

In general, the authors do a good job of presenting the experimental approach and results.  I list below some specific comments that should be addressed.  I think the discussion could do a better job of putting the results on damage and repair rates in the context of other studies.  How do these diatoms compare with other taxa that have been studied and what does that say about their (relative) resistance to PAR and UV inhibition?  One study that is not referenced is that of Sobrino et al. (2007) which examined the responses of the centric diatom, Thalassiosira pseudonana following a similar approach as used in the present study, i.e. comparing the effects of both short-term and long-term shifts in temperature.  Sobrino et al. found that moderate short-term increases in temperature increased damage and repair rates but both rates decreased with long-term acclimation to the same temperature.  It would be interesting for the authors to compare their results with this previous study.  One conceptual difference with the present study is that Sobrino et al., on the basis of exposure-response curves, base their kinetic determinations on an equation that assumes that repair operates at a fixed rate due to an apparent saturation of repair rate at high rates of damage.  This equation is:

$$P = \left( \frac{r}{k} + \frac{r-k}{r} \right) * e^{-kt}$$

Here "P" represents relative rate as a function of time (cf. $P_t/P_0$). This differs from the Kok equation (the author's equation Line 168) which assumes that the contribution of repair to the active pool is proportional to damage.  Which equation is used does have implications for the inferred repair rate which will have different implied units depending on which equation is used, the rate is specific to the pool size of damaged "sites" for the Kok equation but is an absolute rate, fraction of pool repaired with time, for the Sobrino et al. equation.  So the rates can't be directly compared, but the patterns of variation with temperature can.

If further studies are performed on these species, it would be informative to examine different exposures and see if the exposure-response curve is better fit using the model with repair increasing over the full range of exposure (Kok model), or whether repair "saturates" to a fixed rate as for *T. pseudonana*. The latter situation has been generalized into the $E_{max}$ model (Neale et al. 2014), which seems to be broadly applicable to marine phytoplankton.

Specific Comments:

Culture: As mentioned, specify how long cultures were maintained at each temperature before the experiment. Semi-continuous growth – how often were cultures diluted? Growth rates- Methods to determine growth rate (tracking of F0-fluorescence, lines 115-118) more appropriately included with culture conditions section. Specify what was the time interval between T1 and T2. Were multiple determinations made of growth rate for each replicate culture?

Spectra: Line 114-115 discussion of filter transmission is out of place, add to Experimental set up where the cut-off filters are described.

Experimental set up: No information was available on the internet for the radiometer used, please a specific source or details filter type, bandwidth, calibration, etc. Note that a 280 nm cutoff in conjunction with a Xenon lamp means that the samples are being exposed to some irradiance at wavelengths < 290 nm which do not occur under natural solar exposures.

Temperature change: A 10 deg shift could occur in the intertidal benthic environment, but this is not a change that *Skeletonema* is likely to encounter

Chlorophyll fluorescence: It is stated that yield measurements were made on subsamples withdrawn from the treatment tubes. What was the light condition during measurement – I'm guessing it was low or dark. Also, was there a dark adaption period before measurement? If the measurement is not on the sample in treatment irradiance, what is measured is not an effective yield under actinic light, different from what is stated on lines 154-156. Instead the steady-state fluorescence is (or is close to) $F_0'$, minimal fluorescence in the presence of non-photochemical quenching (NPQ) which persists after highlight exposure (depending on the extent of dark adaptation), and the yield is the maximal (or intrinsic) yield. Maximal yield (not dark adapted) will reflect the induction and dissipation of NPQ as well as changes in functional PSII.

Data Analysis: How was "k" estimated from lincomycin treated results – fit to an exponential curve? For both the "k" and "r" fits, statistics should be reported on the standard error of the parameter estimates (available from most non-linear regression routines) and $R^2$ of the fit. In some of the cases of UV exposure, it does not appear as though the Kok equation would give a very good fit as the yield never stabilizes to a steady-state (e.g. results from 15 deg exposures).

In these cases, the uncertainty in parameter estimates will far outweigh the variability associated with replication.

Line 186:  While …
Not a sentence, no verb

Lines 222-225  Not clear what is meant by a "similar pattern".  The decrease in yield in the presence of lincomycin is obviously much greater due to the presence of the inhibitor

Line 229-230 – In the range..
Not a complete sentence

References:

Sobrino, C., and P. J. Neale. 2007. Short-term and long-term effects of temperature on phytoplankton photosynthesis under UVR exposures. J. Phycol. **43:** 426-436.

Neale, P. J., A. L. Pritchard, and R. Ihnacik. 2014. UV effects on the primary productivity of picophytoplankton: biological weighting functions and exposure response curves of *Synechococcus*. Biogeosciences **11:** 2883-2895.

Respectfully submitted,

Patrick Neale
Edgewater, MD

---

## Author Comment (AC1) · 16 Jul 2017

Reviewer #1

This paper presents results from laboratory experiments manipulating the UVR (two levels) and temperature to assess the sensitivity of two diatom species to both factors. The experiment was performed during 120 minutes a single time. The study deals with an interesting topic to phytoplankton ecologists, and tries to clarify a relevant question on the differential photosynthetic responses of the benthic and planktonic species in coastal areas against a scenario of global warming.

Response: We appreciate the comments very much; we would like to make a clarification here that the primary purpose of this study was to test the hypothesis that benthic diatoms have a stronger ability to cope with stressful solar UV radiation under the high temperature regimes that are frequently experienced by benthic species on intertidal flats.

However I find several problems in the manuscript:

The first impression after reading this manuscript is that it is rather long for the type of study done. The topic is interesting, but this is really a snapshot experiment on two hours on two diatoms species. The most suitable presentation of these results would be/could be as a Note and not as a full length paper. On the other hand, this very short-term experiment, with increments of 10 C in temperature, is very unrealistic. Furthermore, a conclusion like this; "the temperature-mediated UV sensitivities might also have implications for phytoplankton in the future warming oceans" seems to me too much speculative.

Response: We agree with the reviewer that the experiment involves short-term light exposure, however, we would argue that in some situations this actually reflects the scenario in the natural environment: the microphytobenthos are often exposed to the coupled stresses of high light and high temperature over a short-term time scale (e.g. during low tide emersion). In addition, we also acclimated both species under different temperatures for at least 5 days before the UV treatment, so that we have data on both short-term and long-term increases of temperature. We believe that this manuscript raises interesting questions that need to be tested more rigorously on a longer time scale under UV radiation, as well as with a broader range of benthic and planktonic species.

For the temperature manipulation, the present manuscript focused on the likely temperature increase on the intertidal flat during low tide periods, rather than mimicking a future scenario of global warming. As measured by Laviale et al., (2015, Environmental Microbiology), the in situ temperature change on the intertidal flat can be greater than 10 °C. Therefore, the simulation of temperature increase in this work is close to what happens in the natural environment. We realized that the last sentence in the abstract might confuse the reviewer that we are dealing with a global warming issue, and this has been deleted in the revision.

My main concern is related to the statistical analysis performed in this study which is not suitable to the experimental design performed and to test the working hypothesis. The authors manipulated two independent factors, so they should do a two-way ANOVA. Also, when authors analysed the effect on variation in the time of the photosynthetic response to light and dim, they should use a RM-ANOVA. Only when they evaluated the temperature effect on the relative UVR inhibition (%), one-way ANOVA is the correct statistical procedure. Moreover, to test their hypothesis, the authors should evaluate the interactive effect UVR and temperature on the two species as well as to quantify the magnitude of these interactive effects. To my impression a wrong test was used. This fatal error determines that the results and discussion must be re-written.

Response: As suggested by the reviewer, we have done this statistical work and found that UV affected both species significantly under all temperature levels except for *Skeletonema sp.* under 35 °C. While the interactive effects of temperature increase and UV were significant for *Skeletonema sp.* over the full range of temperature, and interactive effects were found for *Nitzschia sp.* when temperature increased by 10 °C from 15 or 20 °C; however, no interactive effect was found for the highest temperature (25-35 °C), which we take as strong evidence that *Nitzschia sp.* was relatively resistant to the coupled stresses of high temperature and UV radiation. We have incorporated these results into the revision.

The estimation of the growth rates is confusing. From the description done, it is not easy to understand how was calculated. If I have understood, it was calculated on

fluorescence variation in a 1-hour interval of time, so unit cannot be day; Moreover, I think that the fluorescence is not a good proxy of biomass or abundance, therefore these values did not represent an accurate measurement of growth rates; caution should be taken to discuss this result with those from literature generally obtained from changes of biomass or abundance.

Response: We are sorry that the description about growth rates was not clear. In fact, we measured the fluorescence change over 1 day intervals. As a proxy of biomass or abundance, the most direct estimation is cell counts, POC or *in vivo* chl *a*. Kruskopf and Flynn (New Phytologist, 2005) argued that chlorophyll fluorescence is questionable for biomass estimation of phytoplankton, especially for cultures under nutrient depletion. However, their results actually showed a good correlation between in vivo chl *a* and fluorescence for cultures with relatively lower biomass, <0.25mg chl *a* L$^{-1}$, as the was the case for the cultures in the present study (<0.02mg L$^{-1}$). Consequently, we believe chl *a* is a robust proxy for growth under our experimental set up.

In the results section, there is a lack of precision in the description of the results, making them difficult to understand. The authors should consider remove some of the figures (e.g. Fig. 1 and Fig 2). I think that the figures should be regrouped in two panels, one per each specie, it could benefit the understanding of the Ms. You should present the results in a more synthetic way.

Response: Thanks for the comments, we have moved Fig 1 and Fig 2 into supplementary information. For the arrangement of figures, the primary purpose of our study was to compare species from different niche environments, so we would like to keep the present arrangement with 2 species in one figure, for a better comparison between the two species. We have however made substantial changes to the results section in order describe the data more precisely.

I would like to see the results of the statistical analysis in tables, with the df, F and p values. Likewise, the post hoc results should be presented as part of the figures (lowercase letters).

Response: We have summarized the statistical results as Table A1 and Table A2 in the

supplementary information (also see below), and have also indicated the significance in Fig 4-7 with lowercase letters.

Table A1 The statistical results of RM-ANOVA for the comparison of effective quantum yields under P and PAB at a single temperature level

| species | Temperature type | Temperature level (ºC) | df | F | p |
|---|---|---|---|---|---|
| Skeletonema sp | Acclimated | 15 | 5 | 30.12 | 0.000 |
| | | 20 | 5 | 8.89 | 0.000 |
| | | 25 | 5 | 11.38 | 0.000 |
| | Short term | 25 | 5 | 9.78 | 0.000 |
| | | 30 | 5 | 3.05 | 0.033 |
| | | 35 | 5 | 0.74 | 0.604 |
| Nitzschia sp | Acclimated | 15 | 5 | 38.76 | 0.000 |
| | | 20 | 5 | 10.09 | 0.000 |
| | | 25 | 5 | 13.28 | 0.000 |
| | Short term | 25 | 5 | 11.85 | 0.000 |
| | | 30 | 5 | 9.96 | 0.000 |
| | | 35 | 5 | 5.42 | 0.003 |

Table A2 The statistical results of RM-ANOVA for effective quantum yields during light exposure under different temperature and radiation treatments.

| Species | temperature increase | Factors | df | F | p |
|---|---|---|---|---|---|
| Skeletonema sp | 15-25 | time | 5 | 431.0 | 0.000 |
| | | time*temperature | 5 | 39.43 | 0.000 |
| | | time*light | 5 | 36.17 | 0.000 |
| | | time*temperature*light | 5 | 2.98 | 0.022 |
| | 20-30 | time | 5 | 532.46 | 0.000 |
| | | time*temperature | 5 | 7.85 | 0.000 |
| | | time*light | 5 | 6.39 | 0.000 |
| | | time*temperature*light | 5 | 4.35 | 0.003 |
| | 25-35 | time | 5 | 1127.84 | 0.000 |
| | | time*temperature | 5 | 135.11 | 0.000 |
| | | time*light | 5 | 6.76 | 0.000 |
| | | time*temperature*light | 5 | 2.46 | 0.049 |
| Nitzschia sp | 15-25 | time | 5 | 742.92 | 0.000 |
| | | time*temperature | 5 | 19.46 | 0.000 |
| | | time*light | 5 | 40.5 | 0.000 |
| | | time*temperature*light | 5 | 2.5 | 0.046 |
| | 20-30 | time | 5 | 816.48 | 0.000 |
| | | time*temperature | 5 | 11.12 | 0.000 |

| | time*light | 5 | 16.77 | 0.000 |
|---|---|---|---|---|
| | time*temperature*light | 5 | 3.26 | 0.015 |
| 25-35 | time | 5 | 299.57 | 0.000 |
| | time*temperature | 5 | 4.16 | 0.004 |
| | time*light | 5 | 17.15 | 0.000 |
| | time*temperature*light | 5 | 1.61 | 0.178 |

The authors should pay attention to repetition through the text of terms which was defined in M&M (for instance, photosystem II (PSII), damage rate (k) repair rate (r), Effective quantum yield (y) etc… Likewise, the authors should be consistent with the name of treatments (P exposed not PAR-exposed; UVR vs PAB) through the text; and in figure legends the radiation treatments are written as P or P+UVR whereas in graphs are shown as P and PAB. Finally, the variables should be clearly defined, ( e.g. Relative UV inhibition (%) in figures but in line 159 Relative inhibition (%) etc…).

Response: Thanks for the comments, we have revised the text accordingly throughout the manuscript.

Specific comments

Abstract

It is Ok

Response: No response needed here.

Introduction

Line 85-90. This paragraph might seems repetitive.

Response: We have reworded this paragraph.

Method:

Using the Aquapen fluorometer the authors had to remove 4 ml for each measurement ( I'm assume that the cuvette is 1 cm ), there are 5 measurements in light, 5 in dim plus an initial sample, so in sum about 45ml are needed. How this work if the sample volume had only 35ml?. This needs to be clarified.

Response: The reviewer is correct that the full volume of cuvette is around 4 ml, while during the experiment, we withdrew 2 ml for measurement (which was shown to be adequate by preliminary tests). So a 35 ml sample is enough for the whole experiment.

We have added information to this effect at line 144.

Line 104. both species were inoculated into enriched seawater… It would be necessary to give more details about the culture medium, please.

Response: The medium recipe was Aquil; we have added this information at line 104.

Line 110. Determination of spectra, What do you mean?

Response: Sorry for the confusion, we determined the absorbance spectra of extracted pigments as well as the transmission spectra of cut-off filters. We have reworded this paragraph.

Line 114 . This sentence The cut- off filters were scanned in the same wavelength range against air as a blank. I think it is not the suitable place, because it makes the text confusing.

Response: As suggested by both reviewers, we have reworded this sentence and moved it to line 136-137.

Line 141. A total of 12 tubes (2 species and 2 radiation treatments)…..? The temperature treatments were not made simultaneously? Moreover, how were done the measured of acclimated vs. short-term samples? I can´t understand how the experiment was performed. I hope to be wrong, but seems that the experiment was not a full factorial. In my opinion, the paper would benefit if an illustration of the experimental design would be included.

Response: We have two species under two light treatments (P, PAB) and six temperature treatments, and triplicates for each species so in total we had 2*2*6*3=72 tubes. It is impossible to run all treatments simultaneously, especially for present study to track the kinetics of PSII activity. We then maintained the culture at exponential phase by dilution with fresh medium every day, to keep a stable physiological status, and took samples in the middle of the light period for temperature (2 levels, acclimated, or acclimated+10 ⁰C) and light treatments (P, PAB). We have reworded the appropriate sentences in the M&M and added an illustration (Fig A2) in the supplementary information (shown below).

[Figure]

Line 169. This sentence "where P0 and Pt represent the initial effective quantum yield

and yield at time zero and t (minutes), respectively" is confusing, perhaps is better ……
where P0 and Pt represent the effective quantum yield at time zero and t (minutes),
respectively.

Response: we have reworded this sentence at suggested.

The propagation errors should be applied to calculate the variance of the relative
inhibition UVR (as percentage) as well as the variance in the quotient r:k

Response: We thank the reviewer for this reminder to take into account error
propagation; we have now calculated the variance for relative UV inhibition and the
quotient r:k, and combined these values as error bars in Figure 4, 5 and 6.

Results

Lines 181-186. This paragraph should be removed because the data are not very
informative.

Response: As suggested, we have removed this paragraph and also the related figures.

Line 222-225. I'm sorry, but I don't reach to see what brings to this study the treatments
with antibiotic.

Response: For the study of repair/damage of PSII, lincomycin is often used to block the
repair process, to get a better estimation of rate constant for damage. We have reworded
the sentence at line 127-129, to present the purpose of using the antibiotic more clearly.

This section presents comparisons among different temperatures and radiation
treatments which could not be evaluated by one-way ANOVA, and post hoc analysis,
except to the relative inhibition UVR variable. See above

Response: We have reanalyzed these data by RM-ANOVA, and added p values and F
values in this section, and reworded the sentences as necessary.

Discussion

Line 260-264. This paragraph is very general; I would like to read something about
what is the main contribution of this study.

Response: We have reworded this paragraph as suggested.

The discussion, probably will be modified after addressing the points and questions
related with experimental set-up and statistical analysis.

Response: We have made substantial changes according to the new statistical results.

---

## Author Comment (AC2) · 16 Jul 2017

Reviewer #2

Wu et al. present a study of the photophysiological responses of two diatoms as affected by the temperature during exposure. The responses are observed during short-term exposures to high light (with and without UV) and subsequent recovery periods in low light. By tracking the kinetics of PSII quantum yield during the treatment, inferences can be made about the relative contribution of damage and repair processes to the variations in response between temperature. Additional information can be obtained by exposing the diatoms in the presence of the repair inhibitor lincomycin. This type of approach has been in previous studies of how variation in environmental factors influence inhibition and recovery kinetics, however most studies have focused on a single time scale of treatment, usually on the order of hours to a few days. This study is distinctive in comparing the response to a short-term increase in temperature to responses for cultures acclimated over some growth period to the same temperature. One detail that should be added, however, is how long the acclimated cultures were maintained at their growth temperature before the experiment.

Response: We appreciated the comments very much. The culture was maintained at 3 temperature levels for 5 days before the experiment. For the acclimation time, we have added information at lines 125-127.

In general, the authors do a good job of presenting the experimental approach and results. I list below some specific comments that should be addressed. I think the discussion could do a better job of putting the results on damage and repair rates in the context of other studies. How do these diatoms compare with other taxa that have been studied and what does that say about their (relative) resistance to PAR and UV inhibition? One study that is not referenced is that of Sobrino et al. (2007) which examined the responses of the centric diatom, Thalassiosira pseudonana following a similar approach as used in the present study, i.e. comparing the effects of both short-term and long-term shifts in temperature. Sobrino et al. found that moderate short-term increases in temperature increased damage and repair rates but both rates decreased with long-term acclimation to the same temperature. It would be interesting

for the authors to compare their results with this previous study. One conceptual difference with the present study is that Sobrino et al., on the basis of exposure-response curves, base their kinetic determinations on an equation that assumes that repair operates at a fixed rate due to an apparent saturation of repair rate at high rates of damage. This equation is:

$$P = \left(\frac{r}{k} + \frac{r-k}{r}\right) * e^{-kt}$$

Here "P" represents relative rate as a function of time (cf. Pt/P0). This differs from the Kok equation (the author's equation Line 168) which assumes that the contribution of repair to the active pool is proportional to damage. Which equation is used does have implications for the inferred repair rate which will have different implied units depending on which equation is used, the rate is specific to the pool size of damaged "sites" for the Kok equation but is an absolute rate, fraction of pool repaired with time, for the Sobrino et al. equation. So the rates can't be directly compared, but the patterns of variation with temperature can.

Response: We appreciated the comment, and have read the paper by Sobrino et al., (J. Phycol.2007). One of the main findings in that paper, i.e. "temperature and UVR interact mainly over short (hours) rather than long (days) timescales" offers strong support for present study, since we mimicked the short term increase of temperature likely to be experienced on an intertidal flat, and our data indicated that temperature was a very important factor in influencing microphytobenthos. In addition, the findings of Sobrino et al. on the relationship between BWF and dynamics of repair versus damage was interesting. We have made substantial changes in the discussion. We have not, though, run our data through the Sobrino et al. equation, sticking to the Kok equation for our analysis; as the reviewer states this will not allow absolute rates to be compared but the patterns of variation with temperature will be comparable.

If further studies are performed on these species, it would be informative to examine different exposures and see if the exposure-response curve is better fit using the model with repair increasing over the full range of exposure (Kok model), or whether repair "saturates" to a fixed rate as for *T. pseudonana*. The latter situation has been

generalized into the *E*max model (Neale et al. 2014), which seems to be broadly applicable to marine phytoplankton.

Response: We agree with the reviewer that a comparison with different models is required for future studies. We have additional data on several *Thalassiosira* species that encompass a wide range of size, we hope that we can do a comparison with previous work in our next step.

Specific Comments:

Culture: As mentioned, specify how long cultures were maintained at each temperature before the experiment.

Response: Added as suggested.

Semi-continuous growth – how often were cultures diluted? Growth rates-Methods to determine growth rate (tracking of F0-fluorescence, lines 115-118) more appropriately included with culture conditions section. Specify what was the time interval between T1 and T2. Were multiple determinations made of growth rate for each replicate culture?

Response: The culture was diluted every day with fresh medium. We have added this information at lines 107-108. We have moved the growth rate section into the culture conditions section at lines 111-113, the time interval between T1 and T2 was one day.

Spectra: Line 114-115 discussion of filter transmission is out of place, add to Experimental set up where the cut-off filters are described.

Response: We have moved this sentence to lines 136-137.

Experimental set up: No information was available on the internet for the radiometer used, please a specific source or details filter type, bandwidth, calibration, etc. Note that a 280 nm cutoff in conjunction with a Xenon lamp means that the samples are being exposed to some irradiance at wavelengths < 290 nm which do not occur under natural solar exposures.

Response: The radiometer was produced by a domestic company (http://www.tinel.cn/). The bandwidth of the filters for UVA and UVB were 315-400 nm and 280-315 nm, the radiometer was certified by National Institute of Metrology, China. The sensitivity of this radiometer for UVA and UVB was 0.1 and 0.01 W m$^{-2}$ respectively, and is

somewhat lower than the radiometer that we have used before (ELDONET), but was sensitive enough for the present work. We have added specific source information about this radiometer at line 127.

We agree that the intensity of wavelengths <290nm is negligible at the surface of the Earth. However, because we also want, in future, to run experiments to evaluate the spectral sensitivity of diatoms (and construct biological weighting functions), we used a 280 nm filter here to have a better comparison with future work.

Temperature change: A 10 deg shift could occur in the intertidal benthic environment, but this is not a change that *Skeletonema* is likely to encounter

Response: We agree with the reviewer that a 10 °C rise is unlikely for planktonic species, however, the purpose of our study was to compare species from different niches, so *Skeletonema* here is more likely a reference species. In addition, we have 3 growth temperatures with a 5 °C increase, which could be applicable to coastal phytoplankton.

Chlorophyll fluorescence: It is stated that yield measurements were made on subsamples withdrawn from the treatment tubes. What was the light condition during measurement – I'm guessing it was low or dark. Also, was there a dark adaption period before measurement? If the measurement is not on the sample in treatment irradiance, what is measured is not an effective yield under actinic light, different from what is stated on lines 154-156. Instead the steady-state fluorescence is (or is close to) F0', minimal fluorescence in the presence of nonphotochemical quenching (NPQ) which persists after highlight exposure (depending on the extent of dark adaptation), and the yield is the maximal (or intrinsic) yield. Maximal yield (not dark adapted) will reflect the induction and dissipation of NPQ as well as changes in functional PSII.

Response: The reviewer is correct that the sub-sample experienced a very short-term dark period (<20 seconds) before measurement of chl fluorescence. Strictly speaking, our measurement was not effective yield, nor the dark-adapted value (which requires at least 15 min darkness). However, based on our experience with diatoms that are

exposed to high light/UV, the yield of PSII recovered much slower under darkness than under low light conditions (Wu et al., 2014, J. Photochem. Photobiol. B). So although the value measured in the present work was not perfect, it should be a reasonable operational proxy. To avoid misleading readers, we have reworded the statement at line 159-165.

Data Analysis: How was "k" estimated from lincomycin treated results – fit to an exponential curve? For both the "k" and "r" fits, statistics should be reported on the standard error of the parameter estimates (available from most non-linear regression routines) and R2 of the fit. In some of the cases of UV exposure, it does not appear as though the Kok equation would give a very good fit as the yield never stabilizes to a steady-state (e.g. results from 15 deg exposures). In these cases, the uncertainty in parameter estimates will far outweigh the variability associated with replication.

Response: For the k estimation from the lincomycin treatment, we fitted the lincomycin data into the Kok model with r fixed as zero (when the equation will be $Pt/P0=e^{-kt}$), so it is an exponential curve. We agree with the reviewer that the data fit for some treatments was not good, and resulted in higher standard deviations for some data points. For the quality of the fitting, we summarized r square values in a table as supplementary information (also see below), and hope this could be of help for the reader. We also added related information in the results section.

.

Table A3 R square values for curve fitting with the Kok model for independent replicates of the two species under different temperature and radiation treatments

| Species | Radiation treatment | replicate No. | Temperature treatment (°C) | | | | | |
|---|---|---|---|---|---|---|---|---|
| | | | 15 | 15-25 | 20 | 20-30 | 25 | 25-35 |
| *Skeletonema sp.* | P | 1 | 0.98 | 0.85 | 0.74 | 0.72 | 0.93 | 0.96 |
| | P | 2 | 0.96 | 0.97 | 0.73 | 0.82 | 0.96 | 0.96 |
| | P | 3 | 0.97 | 0.89 | 0.80 | 0.75 | 0.98 | 0.97 |
| | PAB | 1 | 0.91 | 0.94 | 0.92 | 0.97 | 0.97 | 0.99 |
| | PAB | 2 | 0.94 | 0.95 | 0.87 | 0.94 | 0.96 | 0.97 |
| | PAB | 3 | 0.95 | 0.85 | 0.91 | 0.98 | 0.92 | 0.99 |
| *Nitzschi* | P | 1 | 0.77 | 0.84 | 0.78 | 0.96 | 0.87 | 0.98 |
| | P | 2 | 0.74 | 0.89 | 0.75 | 0.93 | 0.82 | 0.96 |

| | | | | | | | |
|---|---|---|---|---|---|---|---|
| P | 3 | 0.74 | 0.84 | 0.73 | 0.86 | 0.88 | 0.90 |
| PAB | 1 | 0.99 | 0.97 | 0.98 | 0.97 | 0.87 | 0.86 |
| PAB | 2 | 0.98 | 0.93 | 0.95 | 0.95 | 0.89 | 0.86 |
| PAB | 3 | 0.97 | 0.96 | 0.96 | 0.97 | 0.93 | 0.88 |

Line 186: While … Not a sentence, no verb

Response: As suggested by reviewer 1, we have deleted this paragraph.

Lines 222-225 Not clear what is meant by a "similar pattern". The decrease in yield in the presence of lincomycin is obviously much greater due to the presence of the inhibitor

Response: Thanks for the comment, we have reworded this sentence.

Line 229-230 – In the range.. Not a complete sentence

Response: Reworded

References:

Sobrino, C., and P. J. Neale. 2007. Short-term and long-term effects of temperature on phytoplankton photosynthesis under UVR exposures. J. Phycol. **43:** 426-436.

Neale, P. J., A. L. Pritchard, and R. Ihnacik. 2014. UV effects on the primary productivity of picophytoplankton: biological weighting functions and exposure response curves of *Synechococcus*. Biogeosciences **11:** 2883-2895.

Response: Thanks for the references, we have cited them in the appropriate places (e.g. line 295-296, 333-334).